# Longevity and Neutralizing Capacity of IgG Antibodies against SARS-CoV-2 Generated by the Application of BNT162b2, AZD1222, Convidecia, Sputnik V, and CoronaVac Vaccines: a Cohort Study in the Mexican Population

Larissa Fernandes-Matano,[a,b] Angel Gustavo Salas-Lais,[c] Concepción Grajales-Muñiz,[d] Mauricio Hernández-Ávila,[e] Yonathan Omar Garfias-Becerra,[f,g] Mario César Rodríguez-Sepúlveda,[e] Carlos Segura-Sánchez,[e] Daniel Montes-Herrera,[c] Denisse Mendoza-Sánchez,[c] Javier Angeles-Martínez,[c] Andrea Santos Coy-Arechavaleta,[c] Julio Elías Alvarado-Yaah,[c] Clara Esperanza Santacruz-Tinoco,[h] Eva Ramón-Gallegos,[b] José Esteban Muñoz-Medina[a]

[a]Coordinación de Calidad de Insumos y Laboratorios Especializados, Instuto Mexicano del Seguro Social, Mexico City, Mexico
[b]Escuela Nacional de Ciencias Biológicas, Programa de Doctorado en Biomedicina y Biotecnología Molecular, Instituto Politécnico Nacional, Mexico City, Mexico
[c]Laboratorio Central de Epidemiología, Instituto Mexicano del Seguro Social, Mexico City, Mexico
[d]Oficina de Vigilancia Epidemiológica del Programa IMSS Bienestar, Mexico City, Mexico
[e]Dirección de Prestaciones Economicas y Sociales, Instituto Mexicano del Seguro Social, Mexico City, Mexico
[f]Research Unit, Institute of Ophthalmology, Conde De Valenciana Foundation, Mexico City, Mexico
[g]Department of Biochemistry, Faculty of Medicine, Universidad Nacional Autónoma de México (National Autonomous University of Mexico), Mexico City, Mexico
[h]Division de Laboratorios Especializados, Instituto Mexicano del Seguro Social, Mexico City, Mexico

Larissa Fernandes-Matano and Angel Gustavo Salas-Lais contributed equally to this article. Author order was determined randomly.

**ABSTRACT** The WHO has approved the use of several vaccines during the COVID-19 pandemic; experience over the last 2 years has indicated that dose demand can only be covered using more than one design. Therefore, having scientific evidence of the performance of the different vaccines applied in a country is highly relevant. In Mexico, 5 vaccines against severe acute respiratory syndrome coronavirus 2 (SARS-CoV-2) were used, allowing a cohort study to analyze the generation of anti-S1/S2 IgG antibodies and anti-RBD antibodies with neutralizing activity at 0, 21, 90, and 180 days after vaccination. Five groups of participants were formed on the basis of the type of vaccine received and were divided on the basis of whether they previously had or did not have COVID-19. After completing the vaccination schedule, the seroprevalence was 95.5, 97.5, 81.0, 95.2, and 90.0% (BNT162b2, AZD1222, Convidecia, Sputnik V, and CoronaVac, respectively). Among the participants without COVID-19 prior to vaccination, the largest amount of antibodies in the 90-day period was observed in the BNT162b2 group, and the amount of antibodies in the Sputnik V group decreased the least over time. Even though the percentages of seroconversion obtained in this study were lower than those currently reported in other parts of the world, the tested vaccines are able, in most cases, to induce a good production of IgG antibodies anti-S1/S2 and neutralizing capacity. The fact that there are people who have not produced antibodies during the study leaves open some questions that must be investigated to avoid the appearance of serious cases of COVID-19.

**IMPORTANCE** Since the start of the vaccination programs against COVID-19 in 2020, it was evident that due to global shortages, the demand for the dose required in Mexico could only be covered by acquiring different vaccines. Therefore, determining the effectiveness of these and the longevity of acquired immunity is extremely important in a scenario where SARS-CoV-2 circulation becomes endemic and booster doses are required periodically. Our data reveal significant differences both in the generation of antibodies as well as in their longevity for the vaccines applied in the country but

Address correspondence to José Esteban Muñoz-Medina, jose.munozm@imss.gob.mx.
The authors declare no conflict of interest.

suggest that, in general, the Mexican population can reach a high capacity to neutralize the virus, therefore, regarding less the variant for which they were designed.

**KEYWORDS** longevity, antibodies, vaccines

The rapid and uncontrollable spread of severe acute respiratory syndrome coronavirus 2 (SARS-CoV-2) infections that has occurred in the world over the last 2 years generated an unprecedented urgency for the development of tools to mitigate the number of infections. As of October 2022, more than 624 million confirmed cases have been recorded, and more than 6.5 million lives have been lost (1); furthermore, more than 12 billion doses of different vaccines against COVID-19 have been administered, i.e., in just over a year after starting vaccination campaigns, 68.4% of the global population has completed a vaccine schedule (2).

Vaccination is the best tool to combat the spread of SARS-CoV-2 infections, reduce the percentage of severe COVID-19 cases, and limit the number of reinfections. As of December 2020, the WHO had approved several vaccines developed for use during the pandemic (3); others are still in the preclinical (196) and clinical (153) stages (4). The vaccines are delivered using various platforms (5), such as mRNA (6, 7), nonreplicative adenovirus vectors (8, 9), and inactivated viruses (10). Among the 4 main coronavirus proteins (nucleocapsid [N], envelope [E], membrane [M], and spike [S]), the N protein, which is related to the transcription and replication of viral genetic material, and the S protein, which is responsible for binding to angiotensin-converting enzyme 2 (ACE2) to enter the host cell through the receptor binding domain (RBD) (11), are the most commonly used antigens for vaccines in development; however, although both are immunogenic, antibodies generated against protein S are correlated with viral neutralization *in vitro* (12).

A separate issue is the production of sufficient doses to meet global demand; although there are a number of vaccine designs, the number of available doses for each vaccine was insufficient throughout 2021. Therefore, the administration of more than 1 design in the same territory has become a common and almost mandatory practice even for first-world countries and is especially evident in developing and underdeveloped countries (13). In Mexico, the Federal Committee for Protection from Sanitary Risks (COFEPRIS) approved the following 5 vaccine designs during the first stages of the National Vaccination Plan (14): BNT162b2 (Pfizer, NY, USA; BioNTech, Mainz, Germany), AZD1222 (AstraZeneca, Cambridge, UK; Oxford University, Oxford, UK), Convidecia (Cansino Biologics, Tianjin, China), Sputnik V (Gamaleya Research Institute, Moscow, Russia), and CoronaVac (Sinovac Biotech Ltd., Beijing, China).

There are several factors that can modify the humoral immune response presented after the administration of these vaccines, for example, the design itself, the vaccination schedule, and the immune status of the host (15). However, 2 of the variables that, in addition to test sensitivity and the selection of antigens used in them, can explain the existing discrepancies between the studies published on this topic (16–19) are the study population and the presence of infection prior to vaccination; notably, evaluations of different vaccines are carried out in different countries and populations, and follow-ups are rarely performed after the day the vaccine is administered.

In this study, we took advantage of the national situation in Mexico, where 5 vaccines against SARS-CoV-2 were used in the same population, and we conducted a cohort study, performing antibody tests beginning on the day of administration of the first dose to determine prior infection among the participants. We focused on determining (i) the amount of IgG antibodies produced targeting the spike protein; (ii) the neutralization capacity of the antibodies produced targeting the RBD; and (iii) the production and decay rates of these antibodies at 0, 21, 90, and 180 days postvaccination to know the best vaccine options for the Mexican population.

## RESULTS

**Demographic analysis.** During the recruitment process, the participants responded to a questionnaire designed to collect personal and general health status data. This information allowed us to determine that 30.8% of the participants who denied having been in contact

**TABLE 1** Demographic and general health data collected from the participants by study group

| Demographic | Study group | | | | |
| --- | --- | --- | --- | --- | --- |
| | BNT162b2 | AZD1222 | Convidecia | Sputnik V | CoronaVac |
| No. of participants recruited | 171 | 209 | 203 | 179 | 214 |
| Previous infection (no. [%]) | | | | | |
| Yes (by IgG and RBD test) | 92 (53.8) | 87 (41.6) | 65 (32.0) | 76 (42.5) | 114 (53.3) |
| Yes (by questionnaire) | 27 (15.8) | 46 (21.9) | 32 (15.8) | 34 (19.0) | 54 (25.2) |
| No (by IgG and RBD test) | 79 (46.2) | 122 (58.4) | 138 (68.0) | 103 (57.5) | 100 (46.7) |
| No (by questionnaire) | 144 (84.2) | 163 (78.1) | 171 (84.2) | 145 (81.0) | 160 (74.8) |
| Asymptomatic (no. [%]) | 65 (38.0) | 41 (19.6) | 33 (16.3) | 42 (23.5) | 60 (28.0) |
| Age range (mean) | 27–76 (55) | 19–78 (50) | 21–60 (39) | 20–60 (49) | 20–63 (36) |
| Sex | | | | | |
| No. (%) female | 113 (66.1) | 135 (64.3) | 150 (73.9) | 104 (58.1) | 123 (57.5) |
| No. (%) male | 58 (33.9) | 75 (35.7) | 53 (26.1) | 75 (41.9) | 91 (42.5) |
| Hypertension (no. [%]) | 17 (9.9) | 48 (22.9) | 10 (4.9) | 39 (21.8) | 15 (7.0) |
| Diabetes (no. [%]) | 10 (5.9) | 36 (17.1) | 6 (3.0) | 27 (15.1) | 8 (3.7) |
| Obesity (no. [%]) | 48 (28.0) | 53 (25.2) | 40 (19.7) | 45 (25.1) | 62 (29.0) |
| HIV (no. [%]) | | 2 (1.0) | | 1 (0.6) | 2 (0.9) |
| Another immunosuppression (no. [%]) | 2 (1.2) | 7 (3.3) | 7 (3.4) | 4 (2.2) | 3 (1.4) |
| Pregnancy (no. [%]) | | | 1 (0.5) | 5 (2.8) | 3 (1.4) |

with SARS-CoV-2 had asymptomatic infections. In total, the percentage of participants who had previously had COVID-19 was 44.5%. Other relevant data were also collected, such as the percentage of different comorbidities among the participants, sex, and average age (Table 1).

**Antibody production in participants with and without prior infection.** Analyzing those participants who presented antibodies in the first sample, that is, participants who had COVID-19 before their first immunization, in the group who presented symptomatic disease, on average, the anti-S1/S2 IgG levels and the neutralizing capacity of anti-RBD antibodies were significantly higher than those in the asymptomatic group. Compared to the participants with COVID-19 prior to vaccination, whether symptomatic or asymptomatic, participants who had not had COVID-19 prior to vaccination had significantly lower IgG antibody levels and a lower neutralizing capacity across the 4 samples collected (Fig. 1).

**Differences in antibody production by sex.** Interestingly, the general analysis by sex showed that at first contact with the antigen (either by natural infection or by vaccination), women produced more anti-S1/S2 IgG antibodies than did men in the group with and without COVID-19 prior to vaccination ($P < 0.05$). However, this difference ceased to be significant when the participants had a second contact with the antigen (Table 2; see also Fig. S1 in the supplemental material).

For anti-RBD neutralizing antibodies, women also showed a higher inhibition rate than did men at first contact with the antigen; however, this difference was only observed in participants without COVID-19 prior to vaccination (50.7 versus 43.7%; $P < 0.05$).

**Seroconversion in participants without COVID-19 prior to vaccination.** Using the data of the participants without COVID-19 prior to vaccination, it was possible to determine the seroconversion rate after the administration of the first and second doses of each vaccine analyzed.

As seen in Fig. 2, the participants who attended the 21-day follow-up, at which time the effectiveness of the first dose of the vaccines could be analyzed, had the highest seroconversion rate among those vaccinated with BNT162b2 (87.8%), followed by those vaccinated with Convidecia (81%), Sputnik V (79.3%), and AZD1222 (71.9%); those vaccinated with CoronaVac had the lowest rate (54.2%).

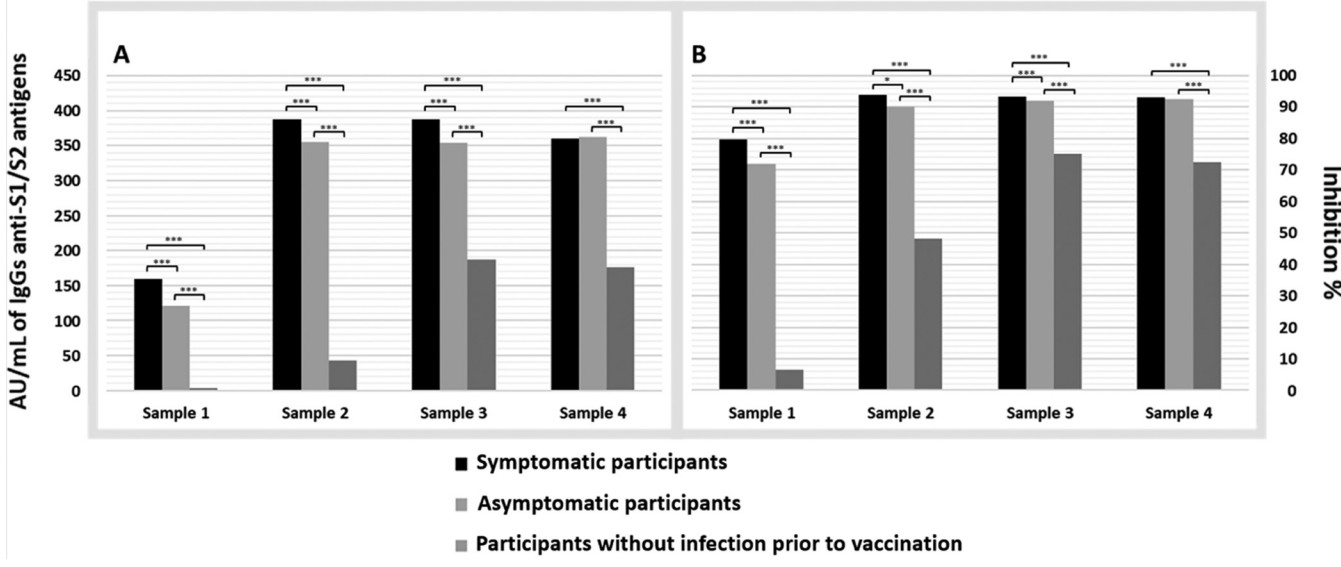

**FIG 1** Average anti-S1/S2 IgG antibodies (A) and the neutralization capacity of anti-RBD antibodies (B) in participants with and without COVID-19 prior to vaccination. *, $P < 0.05$; **, $P < 0.01$; ***, $P < 0.001$.

During the analyzed period, several participants without seroconversion after the first dose, after the second dose, and/or throughout the study (atypical cases) were identified. At day 90, when it was possible to analyze the effectiveness of the complete two-dose schedule, seronegativity was 4.5% in the BNT162b2 group, 2.5% in the AZD1222 group, 4.8% in the Sputnik V group, and 10.0% in the CoronaVac group. With the intention of identifying some aspect in common among these participants, in Table 3, their demographic data, comorbidities, drug therapies, and vaccine received were included.

In general, there were also participants for whom seroconversion was observed after the first dose, although with a low amount of anti-S1/S2 IgG antibodies or a low neutralizing capacity, subsequently returning to values considered seronegative in the following measurements.

**Amount of antibodies in participants immunized with the different vaccines.** Regarding the amount of anti-S1/S2 antibodies, extreme values detected by the interquartile method (IQR) were excluded (Data File S1), and some were analyzed individually (Table 3). The differences found between the groups are shown in Fig. 3.

**TABLE 2** Differences in the production of anti-S1/S2 IgG antibodies and the neutralizing capacity of anti-RBD antibodies between sexes

| Sample | Anti-S1/S2 IgG antibodies production (AU/mL) | | | Percentage inhibition due to the production of antibodies against the RBD portion | | |
|---|---|---|---|---|---|---|
| | Women | Men | P value | Women | Men | P value |
| Participants without infection prior to vaccination | | | | | | |
| Sample 1 | 3.8 | 3.8 | 0.476 | 6.6 | 6.5 | 0.861 |
| Sample 2 | 46.0 | 35.9 | 0.015[a] | 50.7 | 43.7 | 0.045[a] |
| Sample 3 | 185.2 | 189.7 | 0.804 | 74.2 | 76.6 | 0.502 |
| Sample 4 | 177.5 | 171.9 | 0.793 | 71.8 | 73.7 | 0.658 |
| Participants with infection prior to vaccination | | | | | | |
| Sample 1 | 141.5 | 113.3 | 0.006[a] | 77.5 | 73.5 | 0.087 |
| Sample 2 | 383.4 | 374.6 | 0.397 | 96.0 | 95.2 | 0.259 |
| Sample 3 | 380.1 | 367.2 | 0.272 | 96.0 | 95.6 | 0.213 |
| Sample 4 | 371.5 | 355.6 | 0.328 | 95.7 | 95.1 | 0.355 |

[a]Statistically significant differences ($P < 0.05$).

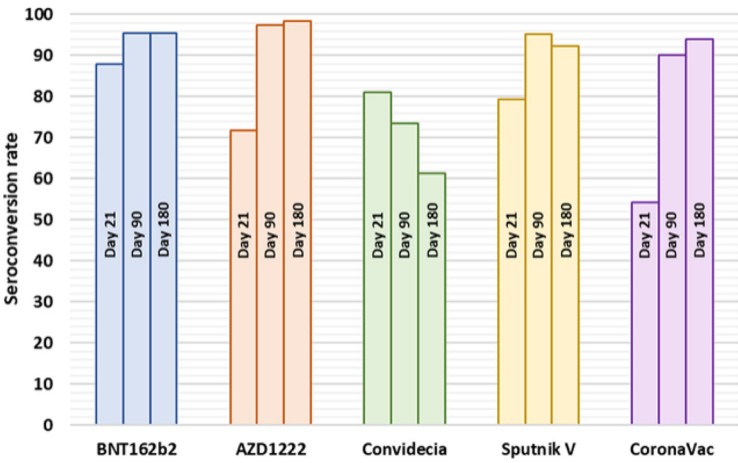

**FIG 2** Seroconversion rates for the groups immunized with the different vaccines at the 3 follow-up time points (days 21, 90, and 180).

At 21 days after the administration of the first dose, in participants without COVID-19 prior to vaccination, CoronaVac induced the lowest production of IgG antibodies (16.8 AU/mL; confidence interval [CI], 12.5 to 21.3), followed by AZD1222 (34.9 AU/mL; CI, 27.8 to 42.0), Sputnik V (41.1 AU/mL; CI, 32.4 to 49.8), Convidecia (55.3 AU/mL; CI, 46.7 to 64.0), and BNT162b2, with the latter generating the highest titers (58.3 AU/mL; CI, 47.0 to 49.6) (Fig. 3B). In participants with COVID-19 prior to vaccination, the production of IgG antibodies was much higher during practically the entire study, and after the administration of the first dose, all of the evaluated vaccines resulted in the maximum production of antibodies detectable by the technique used (400 AU/mL), with the exception of CoronaVac, which presented an average of 305.6 AU/mL (CI, 269.6 to 341.7) (Fig. 3F).

At 90 days, the CoronaVac group continued to have the lowest amount of IgG (Fig. 3C and 3G). However, at the end of the study, participants without COVID-19 prior to vaccination who were administered this vaccine showed the highest amount of IgG antibodies (271.5 AU/mL; CI, 211.1 to 332.0), and those vaccinated with AZD1222 had the lowest amount (94.7 AU/mL; CI, 76.5 to 112.9) (Fig. 3D).

In general, regarding neutralizing antibodies, the results were very similar to those observed for IgG production (Fig. 4), with some exceptions; for example, participants in the BNT162b2 group presented the lowest average inhibition rate after the first dose (92.7%; CI, 89.8 to 95.5; $P < 0.05$), and participants in the Convidecia group presented the lowest at 180 days (50.4%; CI, 41.9 to 58.9).

**Correlation between the production of IgG antibodies and the neutralizing capacity.** To determine if the amount of total IgG antibodies produced by the different vaccines has any impact on the observed neutralizing capacity or if this can occur independently of the amount of anti-S1/S2 IgG present, a correlation analysis was performed. The graphs generated from this analysis show a positive logarithmic correlation between these antibodies (IgG × RBD) (Fig. 5 and 6). To obtain the R2 and the bilateral significance, the outliers were identified and excluded using the Mahalanobis calculation.

At 21 days, in the group vaccinated with BNT162b2, the IgG × RBD ratio was weaker than that in the groups that received other vaccines; in this group, there was a greater dispersion of the points due to high levels of neutralization (Fig. 5A).

In general, at 90 days, the curves shifted to the right, indicating an increase in both anti-S1/S2 IgG antibodies and neutralizing antibodies (quadrant II). This phenomenon occurred in all groups, except for the groups vaccinated with Convidecia and CoronaVac (Fig. 5C and 5E).

By the end of the study, only those vaccinated with BNT162b2 were in quadrants I and II. The other participants, mainly those vaccinated with Convidecia, also occupied quadrant III.

**TABLE 3** Demographic data and general health status of participants with low or no seroconversion

| Vaccine and participant identification no.[a] | Previous infection | IgG/RBD day 21 | IgG/RBD day 90 | IgG/RBD day 180 | Sex/age[c] | Comorbidity/comorbidities | Drug(s) |
|---|---|---|---|---|---|---|---|
| BNT162b2 | | | | | | | |
| 53 | No | 8.7/25.0 | 40.3/63.4 | 9.2/17.4 | F/59 | Diabetes/hypertension | Nifedipine/furosemide/insulin |
| 98 | No | 3.8/6.6 | ND[b] | ND | F/54 | Obesity/diabetes | Metformin |
| 111 | No | 6.0/17.3 | 249.0/93.9 | ND | F/51 | Diabetes/obesity | Omeprazole |
| 142 | No | 7.0/13.4 | 136.0/84.4 | 325.0/79.9 | M/52 | Obesity | ND |
| 79 | No | 41.9/96.7 | 14.3/24.0 | ND | M/55 | Hypertension | Losartan |
| 191 | No | 25.1/48.6 | 5.2/0 | 3.8/11.4 | F/50 | Obesity/hypertension | Levothyroxine |
| AZD1222 | | | | | | | |
| 4 | No | 3.8/22.0 | 117.0/58.6 | ND | F/58 | ND | Loratadine/chlorphenamine/piroxicam |
| 20 | No | 12.9/23.0 | 156.0/92.5 | 43.1/63.5 | F/97 | Obesity | ND |
| 22 | No | 4.2/27.3 | 65.9/56.5 | ND | M/89 | Obesity | ND |
| 39 | No | 3.8/6.4 | 102.0/74.5 | ND | F/61 | ND | ND |
| 52 | No | 7.3/15.5 | 61.2/59.3 | 25.1/30.5 | F/64 | ND | Atorvastatin |
| 79 | No | 3.8/7.9 | 132.0/82.6 | 66.3/66.2 | M/88 | Diabetes | Insulin |
| 112 | No | 11.6/9.4 | 104.0/83.4 | ND | F/83 | Obesity | ND |
| 134 | No | 10.8/14.2 | 121.0/67.2 | 55.5/34.3 | F/69 | Hypertension | Enalapril/sulindac/gabapentin |
| 135 | No | 5.5/18.5 | 28.2/27.9 | 9.1/20.0 | M/90 | Obesity/diabetes/hypertension | Losartan/amlodipine/metformin/allopurinol |
| 148 | No | 10.8/22.4 | 270.0/94.2 | 61.6/57.6 | M/95 | Obesity/hypertension | Enalapril |
| 166 | No | 14.7/27.4 | 310.0/94.4 | 134.0/79.4 | M/94 | Obesity | ND |
| 185 | No | 7.3/9.6 | ND | ND | F/79 | Diabetes | Metformin |
| 208 | No | 10.0/8.6 | 262.0/95.7 | 86.6/87.2 | F/99 | Obesity/hypertension | Losartan |
| 245 | No | 13.1/23.9 | 175.0/85.9 | 48.6/75.1 | F/54 | Immunosuppression | ND |
| 268 | No | 4.7/5.4 | 264.0/95.2 | ND | M/64 | Immunosuppression | ND |
| 272 | No | 3.8/3.3 | 328.0/88.2 | 109.0/64.1 | M/80 | ND | Thyroxine/allopurinol |
| 281 | No | 12.9/16.0 | 132.0/87.0 | 36.1/42.1 | M/77 | ND | ND |
| 290 | No | 5.4/22.3 | 135.0/90.7 | 25.4/53.0 | F/68 | ND | ND |
| 1 | No | NA | 3.8/11.3 | ND | F/58 | ND | ND |
| 137 | No | NA | 3.8/26.2 | ND | M/59 | ND | ND |
| Convidecia | | | | | | | |
| **2** | **No** | **6.1/21.8** | **5.3/10.3** | **3.8/10.1** | **F/42** | **ND** | **ND** |
| **7** | **No** | **11.7/12.4** | **8.9/9.7** | **6.3/8.2** | **F/47** | **Hypertension** | **Losartan** |
| 44 | No | 5.4/15.2 | 400.0/95.7 | 261.0/94.4 | F/31 | ND | ND |
| **46** | **No** | **3.8/4.6** | **3.8/3.5** | **3.8/13.3** | **F/35** | **ND** | **ND** |
| **77** | **No** | **13.8/25.8** | **4.7/7.4** | **3.8/15.6** | **F/35** | **ND** | **ND** |
| **86** | **No** | **9.3/15.1** | **3.8/3.9** | **3.8/14.3** | **F/39** | **ND** | **ND** |
| **92** | **No** | **3.8/9.6** | **3.8/0.0** | **3.8/9.5** | **F/40** | **Obesity** | **ND** |
| **95** | **No** | **3.8/1.0** | **3.8/0.0** | **3.8/11.7** | **M/58** | **Obesity** | **ND** |
| 106 | No | 9.6/3.1 | 5.5/0.0 | 400.0/96.8 | M/37 | ND | Zinc undecylenate |
| 133 | No | 9.8/0.0 | 4.5/16.3 | ND | F/43 | ND | ND |
| **141** | **No** | **14.8/23.1** | **4.4/12.2** | **3.8/8.6** | **F/41** | **Obesity** | **ND** |
| **182** | **No** | **7.3/11.5** | **3.8/0.0** | **3.8/29.2** | **F/42** | **Immunosuppression** | **Simethicone** |

**TABLE 3** (Continued)

| Vaccine and participant identification no.[a] | Previous infection | IgG/RBD day 21 | IgG/RBD day 90 | IgG/RBD day 180 | Sex/age[c] | Comorbidity/comorbidities | Drug(s) |
|---|---|---|---|---|---|---|---|
| 190 | No | 9.5/23.6 | 8.4/19.6 | ND | F/43 | ND | ND |
| **196** | **No** | **3.8/15.6** | **4.2/9.3** | **5.2/11.6** | **F/ 41** | **Immunosuppression** | **Levothyroxine/mycophenolate** |
| 214 | No | 11.9/28.8 | 13.9/33.0 | 12.0/36.6 | F/41 | Obesity | ND |
| 306 | No | 6.9/18.6 | 3.8/11.5 | ND | F/44 | Obesity | ND |
| 20 | No | ND | 9.1/14.3 | ND | M/41 | ND | ND |
| **33** | **No** | **ND** | **11.6/17.6** | **7.52/10.0** | **M/42** | **ND** | **ND** |
| 74 | No | 38.10/50.7 | 9.4/26.0 | ND | M/25 | ND | ND |
| 177 | No | 12.20/35.5 | 8.2/9.0 | 3.80/28.6 | F/28 | ND | Omeprazole/Norfenefrine |
| 197 | No | 24.90/16.1 | 9.2/3.8 | 4.26/8.8 | M/47 | ND | ND |
| **204** | **No** | **ND** | **6.1/24.7** | **4.18/22.6** | **F/53** | **ND** | **ND** |
| 8 | No | 56.2/36.8 | 17.9/27.8 | 3.8/11.3 | M/40 | ND | ND |
| 22 | No | 76.6/57.5 | 43.6/50.1 | 11.4/24.1 | F/45 | Obesity | ND |
| 42 | No | 17.4/77.2 | 15.5/24.3 | 4.6/22.3 | F/46 | ND | Diosmin |
| 101 | No | 48.9/70.2 | 22.2/40.5 | 7.0/26.7 | F/48 | Obesity/diabetes/hypertension | ND |
| 103 | No | 59.3/46.7 | 18.6/32.1 | 5.3/15.1 | F/48 | ND | ND |
| 158 | No | 50.4/41.2 | 34.2/46.0 | 14.1/19.4 | F/39 | ND | ND |
| 183 | No | 17.0/34.9 | 7.5/8.1 | 7.0/17.1 | F/49 | ND | Paracetamol |
| **201** | **No** | **ND** | **ND** | **3.8/15.3** | **M/33** | **ND** | **ND** |
| **208** | **No** | **21.8/11.1** | **ND** | **3.8/0.0** | **M/45** | **Obesity/diabetes/hypertension** | **Insulin/metformin/acetylsalicylic acid** |
| 210 | No | 70.6/49.4 | 37.0/26.2 | 13.5/23.1 | F/44 | ND | Imipramine/pregabalin |
| 212 | No | 59.9/60.5 | 26.7/44.6 | 6.3/11.1 | F/49 | ND | Azulfidine |
| 301 | No | 47.8/35.4 | 19.9/28.6 | 4.7/8.8 | F/32 | ND | ND |
| Sputnik V | | | | | | | |
| 6 | No | 3.8/12.3 | 400.0/96.9 | 337.0/95.0 | F/54 | Obesity | Carbamazepine/ibuprofen/trometamol |
| 8 | No | 6.3/29.3 | 308.0/96.3 | 187.0/92.8 | M/51 | ND | ND |
| 17 | No | 3.8/20.5 | 47.1/41.4 | 22.8/22.7 | M/52 | Diabetes | Insulin |
| **85** | **No** | **3.8/19.3** | **3.8/0.0** | **13.1/17.1** | **M/54** | **ND** | **Metformin** |
| 107 | No | 12.3/27.2 | 154.0/76.6 | ND | F/28 | ND | Multivitamin |
| 122 | No | 5.6/23.8 | 240.0/95.7 | 150.0/87.9 | M/57 | Hypertension | ND |
| **142** | **No** | **4.0/8.6** | **3.8/0.0** | **3.8/14.6** | **M/55** | **Obesity/diabetes** | **Metformin** |
| 148 | No | 9.8/24.7 | 400.0/97.2 | ND | F/21 | ND | ND |
| **149** | **No** | **3.8/0.0** | **3.8/0.0** | **3.8/11.4** | **M/54** | **Immunosuppression** | **Celecoxib/methotrexate** |
| 168 | No | 6.5/20.2 | 353.0/95.7 | 246.0/82.7 | F/52 | Obesity/hypertension | Enalapril/ergotamine/caffeine |
| 197 | No | 10.5/16.5 | 393.0/94.0 | ND | F/42 | ND | ND |
| 203 | No | 13.0/8.2 | 220.0/93.6 | ND | M/23 | ND | ND |
| CoronaVac | | | | | | | |
| 01 | No | 3.8/0.0 | 7.9/27.5 | 277.0/95.4 | M/31 | ND | ND |
| 09 | No | 12.9/29.4 | 26.0/66.3 | 400.0/96.7 | F/31 | ND | ND |
| 10 | No | 8.6/3.3 | 127.0/94.2 | 400.0/96.8 | F/32 | Obesity/hypertension | Amlodipine |
| 23 | No | 5.5/3.5 | 17.0/37.9 | ND | F/36 | Immunosuppression | Fingolimod |
| 25 | No | 13.1/15.3 | 97.7/75.1 | ND | M/38 | ND | ND |
| 43 | No | 3.8/1.7 | 60.5/80.2 | 400.0/96.7 | M/36 | ND | ND |

**TABLE 3** (Continued)

| Vaccine and participant identification no.[a] | Previous infection | IgG/RBD day 21 | IgG/RBD day 90 | IgG/RBD day 180 | Sex/age[c] | Comorbidity/comorbidities | Drug(s) |
|---|---|---|---|---|---|---|---|
| 54 | No | 5.3/12.3 | 41.5/58.5 | 400.0/96.7 | M / 32 | ND | ND |
| 56 | No | 8.3/12.7 | 21.3/40.8 | 400.0/96.9 | F/36 | ND | ND |
| 61 | No | 6.8/15.7 | ND | ND | F/38 | ND | ND |
| 71 | No | 8.1/12.9 | 60.2/58.0 | ND | M / 30 | ND | ND |
| 95 | No | 11.2/13.4 | 46.8/45.8 | 18.8/11.9 | F/38 | ND | ND |
| 128 | No | 7.2/11.6 | 27.1/57.1 | 7.4/38.2 | F/34 | ND | Levothyroxine |
| 144 | No | 9.5/15.8 | 304.0/96.6 | 112.0/95.2 | F/41 | ND | **Olanzapine/clonazepam** |
| **147** | **No** | **3.8/0.0** | **12.8/26.3** | **5.0/16.6** | **F/34** | **ND** | |
| 149 | No | 8.2/4.3 | 78.8/68.5 | 19.1/23.7 | M/39 | Obesity | ND |
| 157 | No | 10.0/14.8 | 43.9/46.5 | 400.0/96.9 | F/37 | ND | ND |
| 169 | No | 3.8/5.4 | 12.8/13.9 | 241.0/95.2 | F/37 | ND | ND |
| 182 | No | 9.6/11.6 | 7.2/22.9 | ND | M/40 | ND | ND |
| 192 | No | 4.4/6.2 | 36.8/73.7 | 400.0/96.7 | F/38 | Obesity | ND |
| 256 | No | 3.8/0.0 | ND | ND | F/– | Diabetes | Levothyroxine |
| 262 | No | 3.8/5.6 | 32.5/50.5 | 384.0/96.7 | M/45 | ND | ND |
| 354 | No | 6.0/2.8 | 30.2/33.5 | 400.0/96.6 | M/23 | ND | Salbutamol |
| 194 | No | ND | 3.8/11.8 | 35.8/42.9 | M / 59 | Diabetes/hypertension | ND |
| 148 | No | 20.5/6.4 | 34.1/35.9 | 11.6/– | M/37 | ND | ND |

[a]The numbers in bold correspond to the participants who never generated antibodies during the duration of the study.
[b]ND, no data; the participant did not attend the sampling, claimed not to have comorbidities, and/or reported not taking medications.
[c]M, male; F, female.

## Participants without infection prior to vaccination

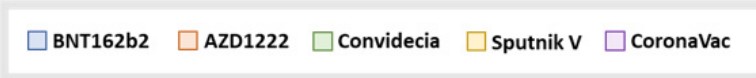

**FIG 3** Anti-S1/S2 IgG antibodies induced by vaccination. Data for the samples taken on day 0 (A), day 21 (B), day 90 (C), and day 180 (D) from participants without COVID-19 prior to vaccination. Data for the samples taken on day 0 (E), 21 (F), 90 (G), and 180 (H) from participants with COVID-19 prior to vaccination. The extreme values shown in the graphs were not considered for comparisons between groups. *, $P < 0.05$; **, $P < 0.01$; ***, $P < 0.001$.

In the group with COVID-19 prior to vaccination (Fig. 6), the logarithmic relationship was maintained only before immunization, that is, with the antibodies generated from natural infection, for all vaccines except for CoronaVac (Fig. 6E).

After the administration of the first dose (21 days), participants with COVID-19 prior to vaccination were in quadrant II. Interestingly, in the BNT162b2 and Sputnik V groups, points were observed in quadrant IV, indicating a low neutralizing capacity of the antibodies generated.

**Production and longevity of antibodies.** For the group without COVID-19 prior to vaccination, for those who received BNT162b2, Sputnik V, and AZD1222, the levels of anti-S1/S2 IgG antibodies increased gradually until peaking at 90 days (2 doses), with means of 331.3, 271.5, and 230.6 AU/mL, respectively. With a cutoff at 180 days, the largest decrease was observed for participants in the AZD1222 group (58.9%), followed by those in the BNT162b2 (38.5%) and Sputnik V (8.1%) groups. For the participants immunized with Convidecia and with CoronaVac, the amount of IgG antibodies at 90 days was much lower than that in participants immunized with the other 3 vaccines; however, after 90 days, a considerable increase of 111.6 and 379.7%, respectively, was observed, possibly due to infection after vaccination because this increase coincides with the increase in cases of COVID-19 in the country (Fig. 7A). Regarding neutralizing antibodies directed against the RBD, although the same tendency is observed, the variations seem to be much more contained and long-lived (Fig. 7C).

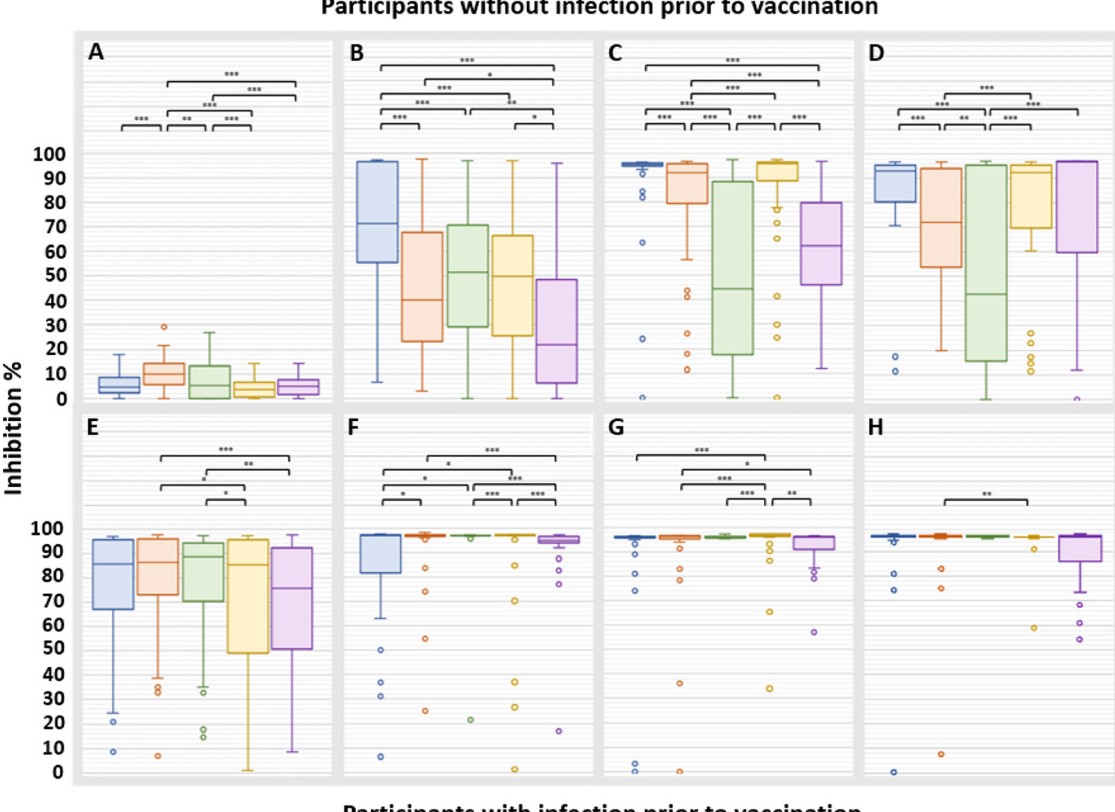

**FIG 4** Neutralizing capacity of anti-RBD antibodies induced by vaccination. Data for the samples taken on day 0 (A), day 21 (B), day 90 (C), and day 180 (D) from participants without COVID-19 prior to vaccination. Data for the samples taken on day 0 (E), 21 (F), 90 (G), and 180 (H) from participants with COVID-19 prior to vaccination. The extreme values shown in the graphs were not considered for comparisons between groups. *, $P < 0.05$; **, $P < 0.01$; ***, $P < 0.001$.

In the group with COVID-19 prior to vaccination, for both anti-S1/S2 IgG antibodies and the neutralizing capacity, for all vaccines, a rapid increase was observed after the administration of the first dose, peaking and remaining high until the end of the study. The only exception, as seen in Fig. 7B, is the average anti-S1/S2 IgG antibodies in the participants in the CoronaVac group, which were much lower than the averages calculated for the participants in the other vaccine groups. The confidence intervals of the percentages mentioned for this analysis are found in Table S1 in the supplemental material.

**Correlation of IgG and RBD with participant age.** Because the National Vaccination Program in Mexico administered vaccines on the basis of age, the participants in each vaccine group belonged to different age groups (Table 1). With the aim of verifying whether this bias had an impact on the results, a correlation analysis was performed between age and IgG production and the neutralization capacity of anti-RBD antibodies.

As seen in Fig. 8, the production of anti-S1/S2 IgG antibodies is only related to the vaccine applied and not to the age of the immunized person. For example, as seen in Fig. 8B, the amount of antibodies in the participants aged 30 to 40 years was smaller in the group vaccinated with CoronaVac than in the participants of this same age group vaccinated with Convidecia, and the amount of antibodies produced by participants aged 50 to 60 years was higher in the group vaccinated with BNT162b2 than in the group vaccinated with Sputnik V. As seen in Fig. 8D, for participants in the age group of 50 to 60 years, those who received AZD1222 had lower levels of antibodies than

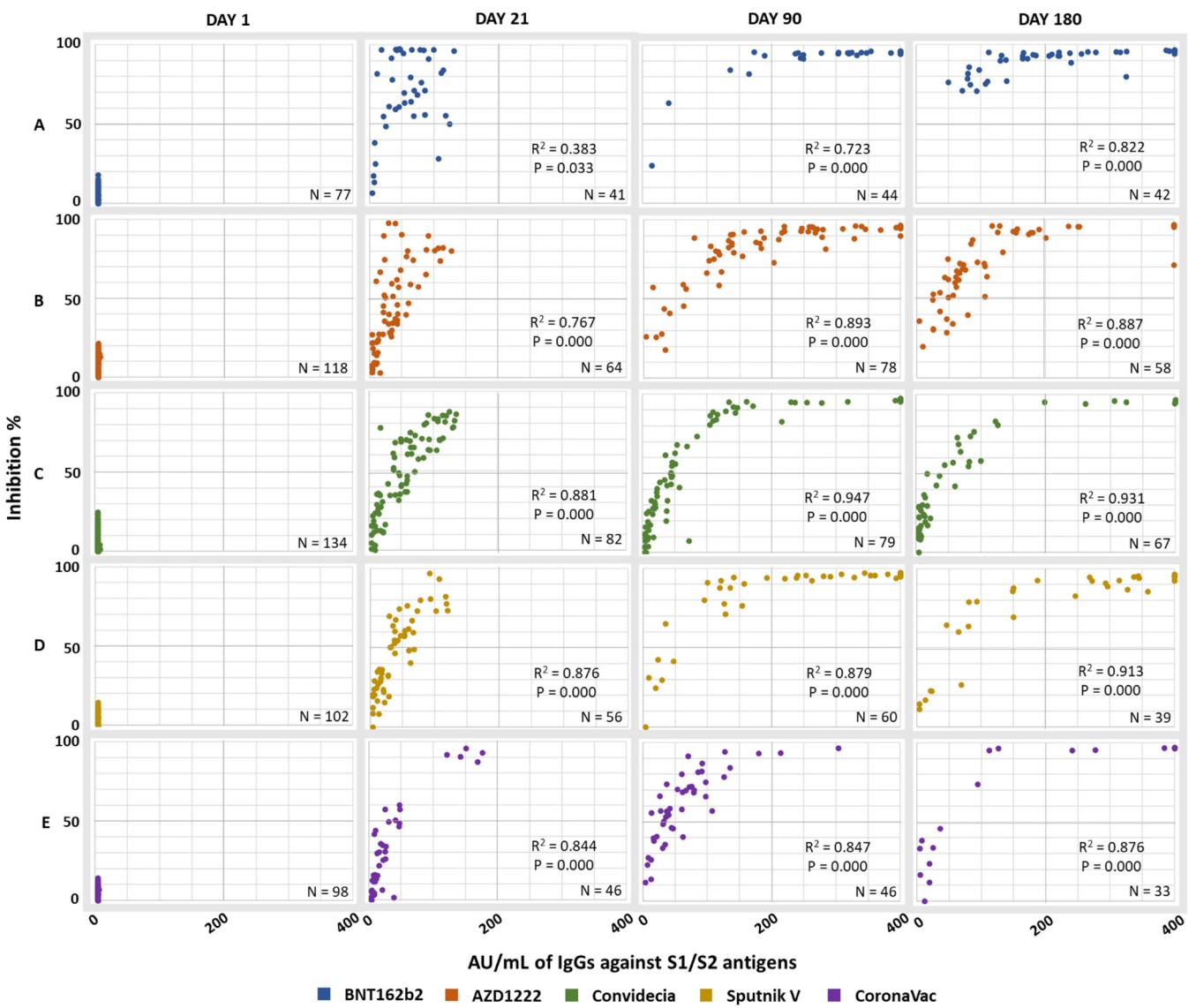

**FIG 5** Correlation between the production of anti-S1/S2 IgG antibodies and the neutralizing capacity of the antibodies directed at the RBD in participants without COVID-19 prior to vaccination. (A) BNT162b2; (B) AZD1222; (C) Convidecia; (D) Sputnik V; (E) CoronaVac.

those who received BNT162b2 or Sputnik V; regarding the neutralizing capacity of anti-RBD antibodies, the same trend was observed (Fig. 8I to P).

## DISCUSSION

One of the main challenges that health systems will have to face in the short term will be to select from the wide range of vaccines against COVID-19 that are currently on the market. Either with the intention of planning for third or fourth doses (boosters) or in the plausible scenario in which vaccination against COVID-19 becomes an annual immunization, knowing the longevity and capacity of the neutralization of antibodies generated by the different vaccines within a specific population is highly relevant information for decision making.

This study analyzed 5 vaccines used in the Mexican population, evaluating the amount of IgG antibodies produced against the spike protein and the neutralizing capacity of the anti-RBD antibodies produced, with a 6-month follow-up with time points at 0, 21, 90, and 180 days postvaccination.

The study design allowed determining, through laboratory techniques, whether the participants had already been exposed to the virus at the time of receiving the first

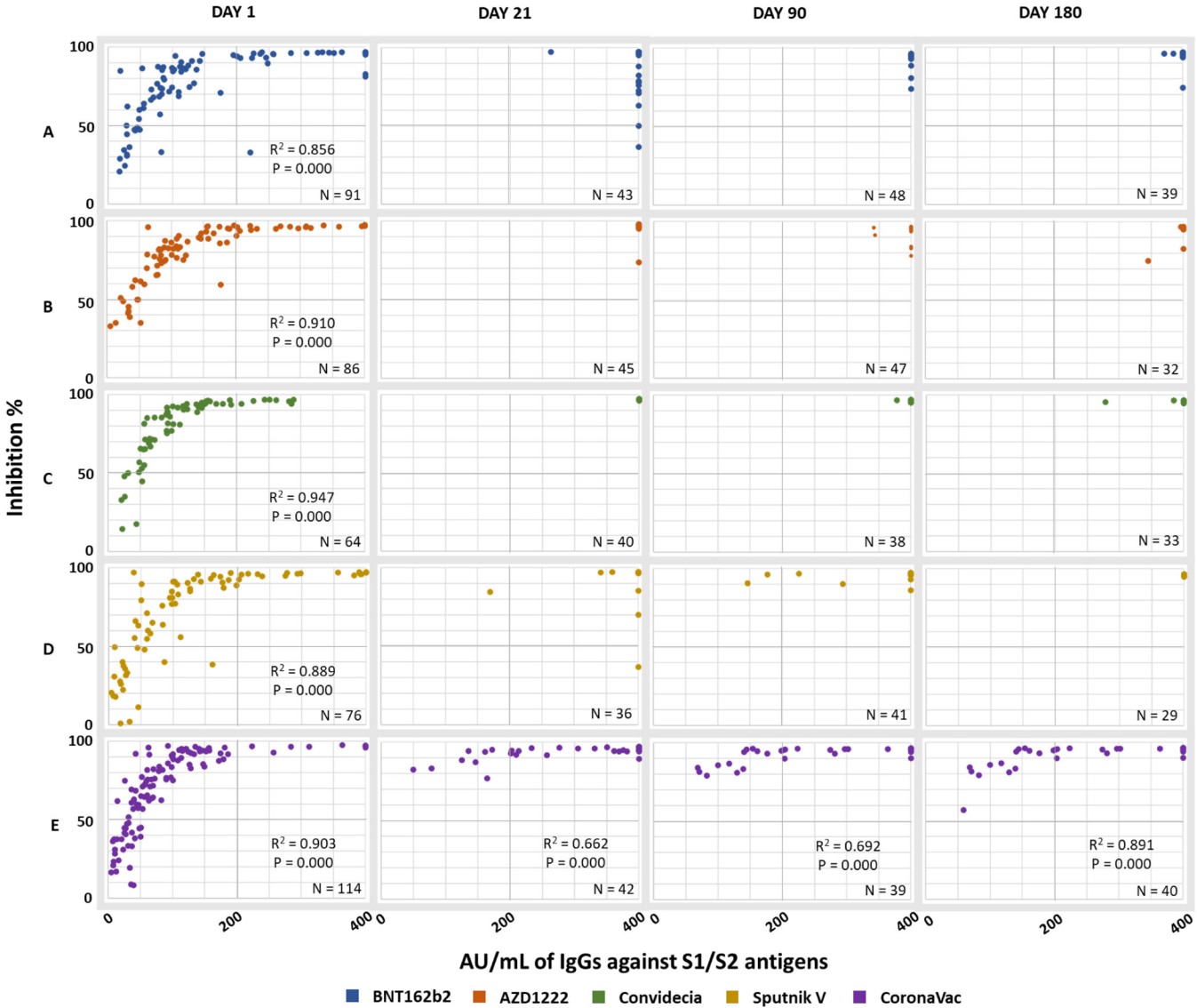

**FIG 6** Correlation between the production of anti-S1/S2 IgG antibodies and the neutralizing capacity of the antibodies directed at the RBD in participants with COVID-19 prior to vaccination. (A) BNT162b2; (B) AZD1222; (C) Convidecia; (D) Sputnik V; (E) CoronaVac.

dose. Additionally, the applied questionnaire allowed us to determine the true percentage of asymptomatic participants (30.8%), but also, and more importantly, it allowed us to separately analyze the results, considering those participants with hybrid immunity (COVID-19 prior to vaccination plus vaccine) and participants with immunity only due to the vaccine. This approach solves the bias present in other studies (20, 21), in which it was not possible to independently determine the humoral response for these 2 scenarios.

At the beginning of the study, the cohort showed a seroconversion rate by natural infection with SARS-CoV-2 of 44.5% (May 2021), which is 11% higher than that reported by Muñoz-Medina et al. (22) from December 2020 in Mexico. Therefore, the real effectiveness of biologics was determined based on 55.5% of the participants being seronegative for SARS-CoV-2. The highest seroconversion rate was observed in participants in the BNT162b2 group after receiving the 1st dose (87.8%), and the lowest seroconversion was observed in participants in the CoronaVac group (54.2%). These results are consistent with those reported in other studies in which mRNA vaccines achieved greater seroconversion than did vaccines based on adenovirus and inactivated whole viruses (23).

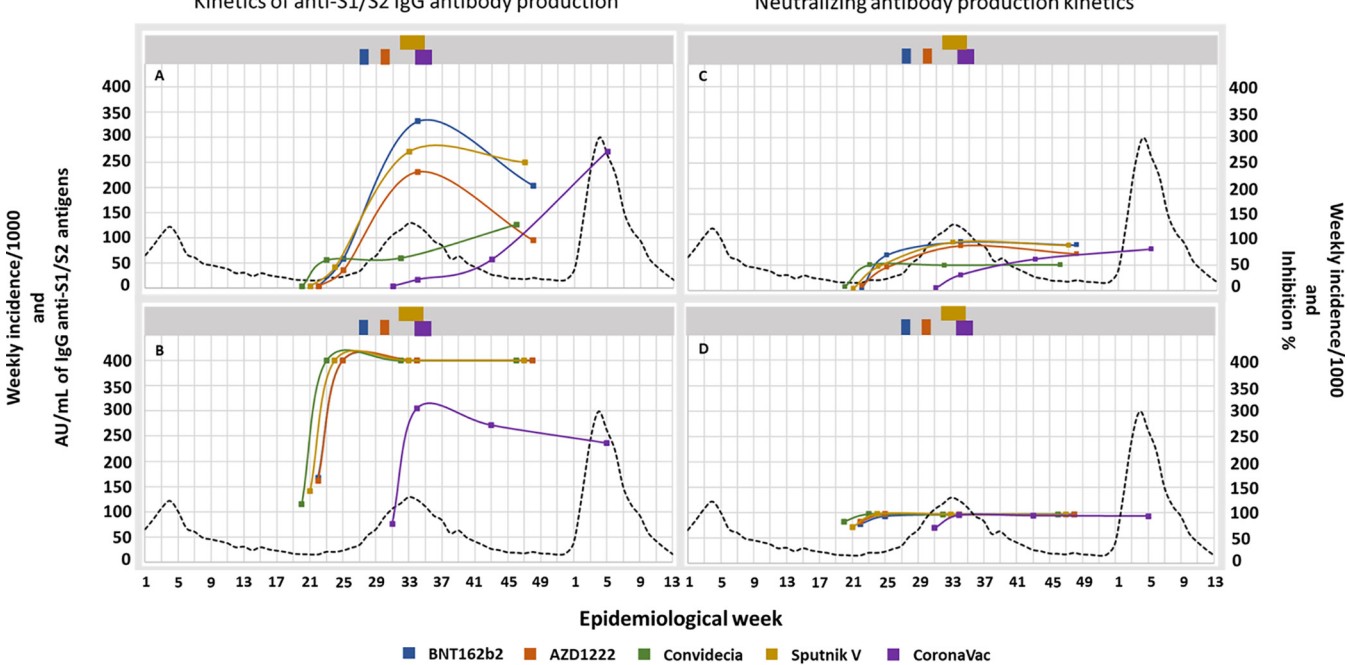

**FIG 7** Analysis of the longevity of the anti-S1/S2 antibodies generated and the neutralizing capacity of the anti-RBD antibodies among the different vaccines. (A) Average anti-S1/S2 antibodies in participants without COVID-19 prior to vaccination. (B) Average anti-S1/S2 antibodies in participants with COVID-19 prior to vaccination. (C) Average inhibition rate of antibodies against the RBD in participants without COVID-19 prior to vaccination. (D) Average inhibition rate of anti-RBD antibodies in participants with COVID-19 prior to vaccination. The upper bars of each graph represent the periods of administration of the second dose. The confidence intervals of the percentages mentioned for this analysis are found in Table S1 in the supplemental material.

After administration of the 2nd dose, the seroconversion rates for participants in the BNT162b2, AZD1222, Sputnik V, and CoronaVac groups were greater than 90% at the end of the evaluation period (180 days after the first dose); in contrast, there was a significantly lower seroconversion in those vaccinated with Convidecia (61.2%). The use of this vaccine has been controversial because at the time of its administration in Mexico (May and June 2021), it was not approved by the FDA (recently approved by the WHO on 19 May 2022) (24); however, it was used because of its availability in the context of the COVID-19 health emergency. Being the only single-dose biological agent, it has a disadvantage compared with the other 4 biologics (2 doses), and this was reflected in the seroconversion percentage (participants without COVID-19 prior to vaccination), peaking at 80.9% at 21 days, similar to that reported by Guzmán-Martínez et al. (25) and Hernández-Bello et al. (26), but decreasing to 73.4% and 61.2% at 90 and 180 days, respectively. This finding is the first report of the long-term efficacy of Convidecia in Mexico and demonstrates the need for a second dose with a compatible biologic to increase efficacy and immunological robustness.

In all vaccine groups, there were participants for whom seroconversion did not occur during the 180 days of follow-up. For participants in the Convidecia and CoronaVac groups, seronegativity after the complete vaccination schedule (19 and 10%, respectively) exceeded the range of 1 to 9% reported by other authors (27, 28).

There are reports of risk factors associated with the persistence of seronegativity after vaccination and/or natural infection, ranging from those related to the evolution of the infection, as for those who are asymptomatic or with mild symptoms, up to a body mass index equal to or greater than 30 (29); however, in our study, no association was found between these factors or with other comorbidities, drug therapies, or demographic data.

In a more detailed analysis of the differences in the generation of anti-S1/S2 and RBD antibodies, in participants with and without comorbidities (see Table S2 in the supplemental material) and in the group without previous COVID-19, there were no significant differences in terms of the anti-S1/S2 IgG antibody response nor in the

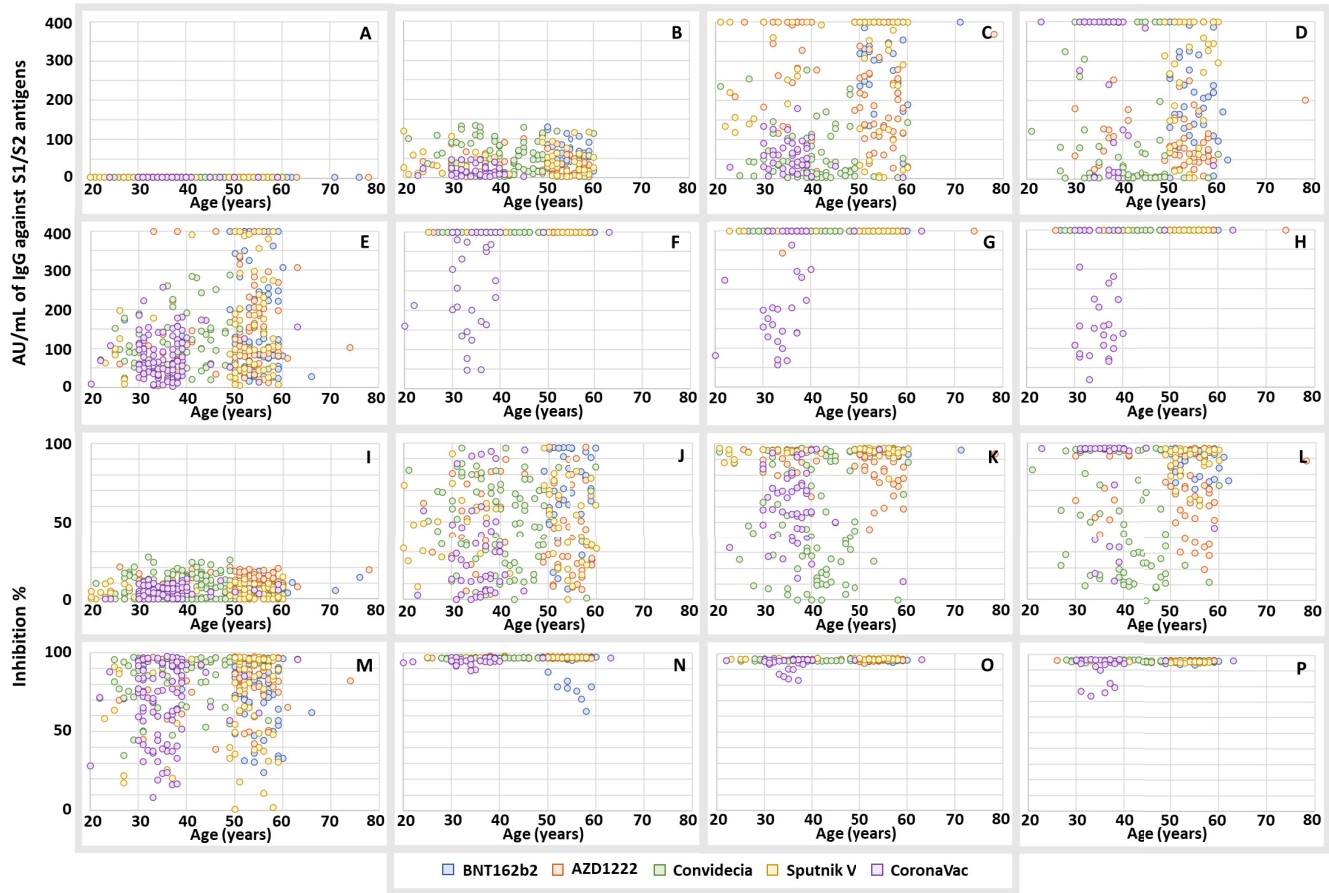

**FIG 8** Correlation between the production of anti-S1/S2 IgG antibodies and the capacity of the anti-RBD antibodies with the participant age. (A to D) Data for the samples taken on days 0, 21, 90, and 180, respectively, from participants without COVID-19 prior to vaccination to analyze the production of IgG anti-S1/S2 antibodies with age. (E to H) Data for the samples taken on days 0, 21, 90, and 180, respectively, from participants with COVID-19 prior to vaccination to analyze the production of IgG anti-S1/S2 antibodies with age. (I to L) Data for the samples taken on days 0, 21, 90, and 180, respectively, from participants without COVID-19 prior to vaccination to analyze the capacity of the anti-RBD antibodies with age. (M to P) Data for the samples taken on days 0, 21, 90, and 180, respectively, from participants with COVID-19 prior to vaccination to analyze the capacity of the anti-RBD antibodies with age.

percentage of inhibition in any of the comorbidities analyzed. However, in the group with previous COVID-19, the people who suffered from arterial hypertension had a higher average of antibodies compared to those who did not suffer from it, a difference that appeared in the first shots and that was no longer significant at 180 days of follow-up.

Lastly, although to a lesser degree, aspects related to the test kits, the vaccine batch administered, the cold chain, and even the administration technique (30) must be considered. The percentage of seronegativity observed is, in reality, the sum of all of these variables.

Because other studies have confirmed the existence of a direct relationship between high titers of IgG antibodies and decreased severity or reinfection (31), it may seem contradictory that in our study, a considerable percentage of participants did not develop antibodies, similar to another study conducted in our country (26). However, as reported by Andreas et al. (32), the lack of IgG antibodies does not necessarily indicate an absence of immunological protection against SARS-CoV-2 because cellular immunity may be active and able to protect against infection.

One variable that can affect the protection provided by vaccines is circulating viral variants. Figure S2 in the supplemental material shows variant of concern (VOC) circulation during the study period, in the area where the samples were obtained. Our data only shows the percentage of neutralizing antibodies versus RBD but not the neutralizing activity against the different variants. A limitation of our study was not performing

neutralization assays against the different variants. Besides, since a large part of the Mexican population was not yet vaccinated during the study period, it is difficult to know if the waves occurred due to the lack of effectiveness of the vaccines used or due to the unvaccinated population.

In a study carried out by Bednarski et al., a reduction in neutralization was observed by Delta and Omicron variants measure by plaque neutralization assays. Even so, although many cases occurred in people vaccinated during the third (Delta) and fourth waves (Omicron BA.1 and BA.2), these cases were generally milder and produced significantly fewer deaths, which indicates that even with the reduction of the neutralizing activity, the applied vaccines limited the evolution to severe stages of the disease (33).

In general, regardless of the biological drug administered, the participants who had been infected before receiving the vaccine had more IgG antibodies and antibodies with neutralizing activity. Even within this same group, those who presented symptoms still had a significantly higher number of antibodies than did asymptomatic individuals. This phenomenon has been widely observed previously (34), indicating a direct correlation between severity and the intensity of the humoral response; however, a less discussed point has been the difference in how long the response lasts. According to our results, the greater number of antibodies in the group of symptomatic participants seems to be only temporary because, at 180 days after vaccination, differences were no longer observed compared with the asymptomatic group.

Regarding the relationship between the production of antibodies between the sexes, to date, there is still controversy regarding whether women are more immunologically reactive than men, either by natural infection or vaccination (35). In our study, women generated a significantly higher amount of IgG antibodies with a neutralizing capacity; however, this was only observed after first contact with the antigen (virus or vaccine), which corresponds to sample 1 for participants with COVID-19 prior to vaccination and to sample 2 for those who had not had prior contact with the virus. If, however, the amount of antibodies generated by women is evaluated after first contact, differences with men would no longer be observed.

When analyzing the general behavior of each vaccine in participants without COVID-19 prior to vaccination, the following 2 profiles of antibody generation can be identified. The first, in which BNT162b2, AZD1222, and Sputnik V are included, is a rapid increase in IgG antibodies accompanied by the presence of antibodies with neutralizing activity, peaking at 90 days and decreasing by 180 days; and the second, which includes Convidecia and CoronaVac, is the generation and maintenance of low levels of antibodies and neutralizing activity, even after completing the vaccination schedule (1 and 2 doses, respectively), with a considerable increase at 180 days.

Regarding the first profile, there was an increase in antibody production at 90 days due to the administration of the second dose and possible infections caused by the Delta variant wave present at that time. At 180 days, the Delta wave did not appear to trigger a response; therefore, the observed decrease in antibodies was expected (36). This is because not all vaccine-induced plasmablasts are maintained as antibody-producing plasma cells (37).

In addition, with our results, we were able to corroborate that compared to the adenoviral platform, BNT162b2 produces a larger amount of antibodies (in this case, Sputnik V), and in turn, Sputnik V was shown to induce a greater amount of antibodies than was AZD1222. The rapid decrease in antibodies that was observed in the participants who received AZD1222 was also reported in other studies (38, 39). Sputnik V showed the greatest stability in the production of anti-S1/S2 IgG antibodies and anti-RBD neutralizing antibodies between 90 and 180 days after the 1st dose, as reported by Sánchez et al. (40). This stability can be attributed to the vaccination scheme implemented by the Mexican government for this biologic, for which the interval between the 1st and 2nd doses was 80 to 96 days. There is now evidence that the 1st dose of this biologic (currently known as Sputnik light) can be applied as another vaccine alternative (41).

Regarding the second profile, the increase in antibodies that occurred after receiving the complete vaccination schedule in the participants of Convidecia and CoronaVac can only be explained by an infection caused by the Delta and Omicron waves, respectively.

Regarding CoronaVac, the antibody profile generated in the participants without COVID-19 prior to vaccination contrasted strongly with that generated in the participants with COVID-19 prior to vaccination; this result is interesting because the 2 groups were equally exposed to infection during the Omicron wave. Only in the first case did the antibodies increase, which would indicate that in only the participants who had new contact with the antigen or in only those in that group was the infection successful. This could be due to the low neutralizing capacity of the antibodies generated thus far in the participants without COVID-19 prior to infection (62.4%) compared to that in those with hybrid immunity (94.5%) as reported by Zhao et al. (42). However, in this study, it was not possible to confirm that the number of infections was lower in a specific group. The potential exposure to and infection by SARS-CoV-2 in participants without COVID-19 prior to vaccination, after completing the CoronaVac vaccination schedule, coincides with our results obtained in the evaluation of IgG antibodies against the nucleocapsid, i.e., the amount produced was significantly higher on day 180 (168.8 AU) than on day 0 (38.54 AU), 21 (52.83 AU), and 90 (70.73 AU) (in the process of writing). Anti-N IgG antibodies against SARS-CoV-2 generated in people vaccinated by inactivated viruses, e.g., CoronaVac, are present at low levels (43), and these only increase when there is a postvaccination infection (44).

"Hybrid immunity," as vaccination after natural infection by SARS-CoV-2 has been called, has been shown to strengthen and prolong the humoral immune response against SARS-CoV-2 (45) because the neutralizing antibodies generated naturally by the infection can persist up to 9 months later with 66.1% neutralizing activity as reported by Shim et al. (46). In our study, we had an extensive group of participants with COVID-19 prior to vaccination, meaning that they were in contact with the predominant variants of SARS-CoV-2 during the first waves of the pandemic in Mexico (47). In the group with hybrid immunity, unlike the findings reported by other studies (48, 49), antibody generation and neutralizing activity were observed, reaching maximum detectable values by the kits (400 AU/mL and almost 100% inhibition) after the first dose and remaining high until the end of the study (180 days). This trend was present even in the Convidecia group, which had a single-dose schedule. Although some studies report that the production of anti-S IgG antibodies decreases rapidly after the administration of CoronaVac, regardless of whether individuals had COVID-19 prior to vaccination (50), the levels of neutralizing antibodies in our group remained above 90% throughout the study.

The correlation analysis between the amount of IgG produced and the neutralization capacity showed that, contrary to what was expected, these follow a logarithmic relationship and not a linear relationship, which means that, even at the point where a person produces few anti-S1/S2 IgG antibodies, there will not necessarily be a low capacity to neutralize the virus. A clear example that even with a low amount of anti-S1/S2 IgG antibodies, high levels of neutralization can be achieved, is what is observed in the group vaccinated with BNT162b2 after the first immunization. In this group, the logarithmic ratio was lower than that in the groups vaccinated with other brands because more people reached high levels of neutralization despite the low amount of anti-S1/S2 IgG antibodies they produced in general.

This finding becomes relevant to the interpretation of other works in which population immunity is measured based on kits directed against protein S and not specifically on the ability to neutralize the virus (51).

Finally, like other studies on this topic, ours also has limitations. The first and most important was that due to economic reasons, it was not possible to determine antibodies against protein N in all groups, so we could not identify the participants who had a postvaccination infection. This generated uncertainty about the production kinetics and longevity of the antibodies, since the reported trends may include biases derived

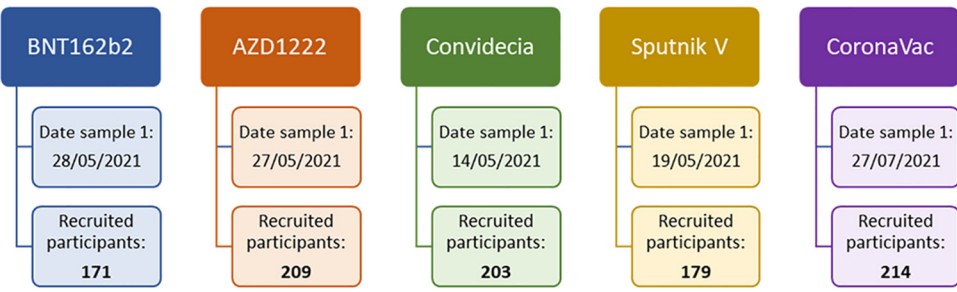

**FIG 9** Participants were recruited by vaccination group and start date of follow-up.

from the exposure of the participants to SARS-CoV-2 during the peaks of infection in Mexico and the appearance of new variants.

Another limitation was that, although we detected a high production of antibodies derived from vaccination with the biologics studied, with our data it is not possible to determine whether the circulating variants of SARS-CoV-2 escape the immune response generated in the vaccinated population. And finally, the dropout of certain participants during the study reduced the number of samples analyzed at each follow-up, which directly affected the statistical power of the observed differences and correlations.

In conclusion, our results showed that, from the first immunization, the analyzed vaccines caused different levels of seroconversion in the population (without previous infection), whose percentages varied from 54.2 to 87.8%, with CoronaVac being the vaccine with the lowest and BNT162b2 with the highest percentage of seroconversion. Since everything indicates that SARS-CoV-2 will become a seasonal virus, BNT162b2 should be the vaccine of choice for the first immunization. After the second immunization, all of the vaccines caused seroconversion levels above 90% (except Convidecia, which had a single-dose schedule). Despite the fact that the seroconversion percentages obtained in this study were lower than those reported in other parts of the world, given the current global scenario, where a good part of the population has already had a natural SARS-CoV-2 infection, the application of any of the biologicals analyzed in this study manages, in most cases, to induce a good production of anti-S1/S2 and neutralizing IgG antibodies, so they could be an option as a booster dose, as long as care is taken to ensure compatibility with the biological applied initially.

The fact that there is a percentage of the population in which there was no production of antibodies during the 180 days of the study leaves open some questions about the effectiveness of the vaccines themselves, the protection generated by the cellular immune response, the form of application, the cold network used to store them, and the associated human factor. All of these factors must be taken care of and investigated to avoid the appearance of serious cases of COVID-19.

## MATERIALS AND METHODS

**Study design.** With the intention of evaluating the humoral immune response generated by the administration of 5 different vaccines against COVID-19 approved in Mexico, a prospective cohort study was conducted; participants were recruited in some of the mass vaccination centers located in the State of Mexico (Mexico). The candidates were divided into 5 groups on the basis of the vaccine that they received, with 171 receiving BNT162b2 (date sample 1, 28 May 2021), 209 receiving AZD1222 (date sample 1, 27 May 2021), 203 receiving Convidecia (date sample 1, 14 May 2021), 179 receiving Sputnik V (date sample 1, 19 May 2021), and 214 receiving CoronaVac (date sample 1, 27 July 2021) (Fig. 9).

The follow-up for all groups was the same and consisted of obtaining 4 samples of peripheral blood, starting on the day of administration of the first dose, with the purpose of dividing the participants on the basis of the occurrence or not of a previous infection, and subsequently at 21, 90, and 180 days. For each sample, chemiluminescence immunoassay (CLIA) was performed to assess the levels of anti-S1/S2 IgG antibodies, and a neutralization enzyme-linked immunosorbent assay (ELISA) was used to measure the neutralizing capacity of the anti-RBD antibodies.

The study was registered for evaluation by the IMSS National Scientific Research Committee (registration number R-2022-785-037) and was carried out in accordance with the Declaration of Helsinki. At the beginning of the study, all participants signed informed consent forms and completed a questionnaire regarding personal data and their general health status.

Blood samples were obtained in 5-mL BD Vacutainer tubes with separator gel. After collection, all samples were centrifuged (10 min/3,500 rpm) and sent, under refrigeration conditions, to the Central Epidemiology Laboratory (LCE) of the IMSS through institutional mail and triple-packaged as a category B biological substance (UN 3373) following the recommendations of the International Air Transport Association (IATA) (52).

**Procedures. (i) Detection of IgG antibodies against the S1/S2 antigens of SARS-CoV-2.** The detection of IgG antibodies was performed using 200 $\mu$L of serum and a LIAISON SARS-CoV-2 S1/S2 IgG kit (DiaSorin, Saluggia, Italy; catalog number 311450) (53); the tests were conducted in Liaison XL equipment following the manufacturer's instructions. Negative and positive controls were used to validate the results. The cutoff values were as follows: negative < 15 AU/mL and positive $\geq$ 15 AU/mL.

**(ii) Neutralizing activity of the antibodies against the RBD antigen of SARS-CoV-2.** The neutralizing activity of the anti-RBD antibodies was assessed using an ELISA SARS-CoV-2 surrogate virus neutralization test kit (GenScript, NJ, USA; catalog number L00847) (54). The kit contains the cell surface receptor ACE2 immobilized on 96-well plates and includes an analog to the horseradish peroxidase (HRP)-labeled receptor binding domain (HRP-RBD), which was previously incubated with the serum to be tested. The neutralizing antibodies in the serum bind to HRP-RBD and block its interaction with the cellular receptor ACE2. The tests were performed following the manufacturer's recommendations, and negative and positive controls were used to validate the results. The cutoff values were as follows: negative < 30% inhibition and positive $\geq$ 30% inhibition.

**Statistical analysis.** Descriptive statistics were used to evaluate frequency and percentage reports; these were calculated with their respective 95% confidence intervals. The outliers were identified using the IQR (interquartile range) and Mahalanobis calculation, depending on the conditions. The $\chi^2$ and Kruskal-Wallis tests were used to compare categorical variables. To cross the factors with the dependent numerical variables, one-way analysis of variance (ANOVA), multifactorial analysis, multivariate analysis of variance (MANOVA), analysis of covariance (ANCOVA), multivariate analysis of covariance (MANCOVA), or the Mann-Whitney U test was used as appropriate. Correlation analyses were performed using the Pearson or Spearman test. Values of $P < 0.05$ were considered significant. IBM SPSS Statistics 24.0 was used for the analyses.

**Data availability.** All data are available in the main text or the supplementary materials.

## SUPPLEMENTAL MATERIAL

Supplemental material is available online only.
**SUPPLEMENTAL FILE 1**, XLSX file, 0.2 MB.
**SUPPLEMENTAL FILE 2**, PDF file, 0.4 MB.

## ACKNOWLEDGMENTS

We want to thank the participation of the Auxiliary Coordinators of Medical Care, Directors of Medical Units, and Heads of Laboratory of the IMSS for their support during the follow-ups.

A.G.S.-L. is the recipient of postdoctoral fellowships from CONACyT (408350).

Conceptualization, L.F.-M., A.G.S.-L., C.G.-M., and J.E.M.-M.; Data curation, L.F.-M., A.G.S.-L., D.M.-H., and J.A.-M.; Formal analysis, L.F.-M., A.G.S.-L., J.E.M.-M., E.R.-G., and Y.O.G.-B.; Investigation, L.F.-M., A.G.S.-L., and J.E.M.-M.; Methodology, L.F.-M., A.G.S.-L., J.E.M.-M., D.M.-S., D.M.-H., J.A.-M., A.S.C.-A., and J.E.A.-Y.; Project administration; L.F.-M., A.G.S.-L., J.E.M.-M., D.M.-S., D.M.-H., and J.A.-M.; Resources, Y.O.G.-B., M.H.-A., M.C.R.-S., C.S.-S., J.E.A.-Y., C.G.-M., and C.E.S.-T.; Supervision, L.F.-M., A.G.S.-L., J.E.M.-M., A.S.C.-A., E.R.-G., M.H.-A., M.C.R.-S., C.S.-S., and C.E.S.-T.; Visualization, L.F.-M., A.G.S.-L., and J.E.M.-M.; Writing – original draft, L.F.-M., A.G.S.-L., and J.E.M.-M.; Writing – review and editing, L.F.-M., A.G.S.-L., C.G.-M., M.H.-A., Y.O.G.-B., M.C.R.-S., C.S.-S., D.M.-H., D.M.-S., J.A.-M., A.S.C.-A., J.E.A.-Y., C.E.S.-T., E.R.-G., and J.E.M.-M.

We declare no conflict of interest.

The funders had no role in the design of the study; in the collection, analyses, or interpretation of data; in the writing of the manuscript, and in the decision to publish the results.

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
