## [Reviewer comments · Microbiology Spectrum]

Microbiology Spectrum

Longevity and neutralizing capacity of IgG antibodies against SARS-CoV-2 generated by the application of BNT162b2, AZD1222, Convidecia, Sputnik V and CoronaVac vaccines: A cohort study in Mexican population.

Larissa Fernandes-Matano, Angel Salas-Lais, Concepción Grajales Muñiz, Mauricio Hernandez-Avila, Yonathan Omar Garfias-Becerra, Mario César Rodríguez-Sepúlveda, Carlos Segura-Sanchez, Daniel Montes-Herrera, Denisse Mendoza-Sánchez, Javier Ángeles-Martínez, Andrea Santos Coy-Arechavaleta, Julio Elías Alvarado-Yaah, Clara Santacruz-Tinoco, Eva Ramón-Gallegos, and Jose Esteban Muñoz-Medina

Corresponding Author(s): Jose Esteban Muñoz-Medina, Instituto Mexicano del Seguro Social

Review Timeline:

Submission Date:	June 30, 2022
Editorial Decision:	October 23, 2022
Revision Received:	November 10, 2022
Accepted:	December 6, 2022

Editor: Alison Sinclair

Reviewer(s): Disclosure of reviewer identity is with reference to reviewer comments included in decision letter(s). The following individuals involved in review of your submission have agreed to reveal their identity: Pragma D. Yadav (Reviewer #1)

Transaction Report:

DOI: <https://doi.org/10.1128/spectrum.02376-22>

October 23, 2022

Dr. Jose Esteban Muñoz-Medina
Instituto Mexicano del Seguro Social
Laboratorio Central de Epidemiología
Mexico
Mexico

Re: Spectrum02376-22 (Longevity and neutralizing capacity of IgG antibodies against SARS-CoV-2 generated by the application of BNT162b2, AZD1222, Convidecia, Sputnik V and CoronaVac vaccines: A cohort study in Mexican population.)

Dear Dr. Jose Esteban Muñoz-Medina:

Link Not Available

Sincerely,

Alison Sinclair

Journals Department
Reviewer comments:

Reviewer #1 (Comments for the Author):

The study Title: Longevity and neutralizing capacity of IgG antibodies against SARS- 2 CoV-2 generated by the application of BNT162b2, AZD1222, Convidecia, Sputnik V, and CoronaVac vaccines: A cohort study in Mexican population, Authored by Larissa Fernandes-Matano et al have described the vaccine status response situation in Mexico, where 5 vaccines against SARS-CoV-2 were used in the same population, and conducted a cohort study, performing antibody tests beginning on the day of administration of the first dose to determine prior infection among the participants. They have focused on determining i) the number of IgG antibodies produced targeting the spike protein; ii) the neutralization capacity of the antibodies produced targeting the RBD; and iii) the production and decay rates of these antibodies at 0, 21, 90, and 180 days postvaccination, to

know the best vaccine options for the Mexican population.

Comments

Study is designed well and gives good value to the current scenario. It would be useful and provide more value if this data can be correlated with the presence of a Variant of concerns or variants present during the study time. VOCs do affect the neutralization capacity of vaccine and their efficacy so if it's possible then incorporate or use other published data to describe the possible circulating VOCs.

Also how you have taken care of the individuals who were infected earlier with COVID and vaccinated or vaccinated and later got infected? Did this impact the outcome?

Did study was followed post-180 days to see the impact of breakthrough, re-infection or other impacts

Was booster dose implemented in a later phase

How many breakthrough cases were reported during the study

Those did not develop antibody, have breakthroughs in follow-up period of 6 months or not

Was comorbid group show any significant difference with the healthy group for antibody or NAb difference

Was this neutralization enzyme-linked immunosorbent assay (ELISA) specific to measure the neutralizing capacity of the anti-RBD antibodies for any specific VOCs??

Limitation of the study should be expressed

Reviewer #2 (Comments for the Author):

The work by Fernandes-Matano, Salas-Lais et al. explore the differences at the humoral response level, specifically IgG, in a cohort in Mexico after vaccination with five different vaccines approved in the country against SARS-CoV-2. Several countries in need of controlling SARS-CoV-2 adopted a strategy of vaccinating the population with multiple vaccines based on availability. Therefore, understanding the differences among different vaccine platforms is of interest in other regions and, overall, for pandemic preparedness. The information provided in this article and the samples used for the experiments are valuable. The experimental design is mostly accurate, and the science is sound. However, there are specific flaws in the experimental design mainly associated with ruling out the presence of a natural infection during the experiment that weak the interpretation of the results. The manuscript will considerably benefit if authors decipher whether natural infections happened during the experiment. Regardless of that, the information is valuable and exciting for the readers of the journals. My specific comments are listed below:

Major comments

A significant concern is a potential role that natural SARS-CoV-2 infection could have in everyone during the sample collection (Day 0-180). Is the information regarding close contact with SARS-CoV-2-positive people or clinical signs associated with SARS-CoV-2 during the sample collection available? How was the VOC circulation during the experiment timeline? Could this be a variable? Besides CoronaVac, the rest of the vaccines are Spike-based. Therefore, antibodies against N are a tool to differentiate natural infection and vaccination. That information will strongly support the idea of a lack of disease during the sample collection period. Anti-N antibodies are a piece of information already available to the authors based on the discussion.

An increasing body of literature demonstrates that biological sex plays a role in the immune response against vaccination and pathogenesis against respiratory viruses such as influenza or SARS-CoV-2. The authors successfully analyzed potential sex differences, finding such differences in the cohort analyzed (Line 121-23), which is highly informative in this reviewer's opinion. However, the authors failed to show these results appropriately, and just a table is in place. I strongly encourage the authors to present a figure based on sex differences.

Correlations with age were carried out. However, the biological sex was not analyzed.

Please discuss more in detail about potential explanations for the differences in the correlation between S1/S2 IgG vs. neutralization capacity between no prior covid and prior covid positive individuals.

Conclusions should state better the categorization of the vaccines based on the experiments carried out

Minor comments

Line 61-63: Please update the number of cases, deaths, and vaccines administered until September 2022 at least

Timeline of sample collection: Please specify specific numbers of individuals and the date of sample collection in materials and methods

Fig 1 will benefit from the use of a more precise nomenclature. Along the papers, the different samples are denoted by the day

of collection, which I think should be the way to go in this figure.

Fig 3: Please specify each day on top or under each panel, so it's clear for the readers

Line 161: Please check the figures described. It should be 3C and 3G.

Fig 5 and 6. Please specify the name of each vaccine instead of A, B, C, etc.

Fig7: Please check the figure legend for C and D (is the same?)

Fig 8 and 9 should be combined.

Staff Comments:

Preparing Revision Guidelines

Please return the manuscript within 60 days; if you cannot complete the modification within this time period, please contact me. If you do not wish to modify the manuscript and prefer to submit it to another journal, please notify me of your decision immediately so that the manuscript may be formally withdrawn from consideration by Microbiology Spectrum.

The work by Fernandes-Matano, Salas-Lais et al. explore the differences at the humoral response level, specifically IgG, in a cohort in Mexico after vaccination with five different vaccines approved in the country against SARS-CoV-2. Several countries in need of controlling SARS-CoV-2 adopted a strategy of vaccinating the population with multiple vaccines based on availability. Therefore, understanding the differences among different vaccine platforms is of interest in other regions and, overall, for pandemic preparedness. The information provided in this article and the samples used for the experiments are valuable. The experimental design is mostly accurate, and the science is sound. However, there are specific flaws in the experimental design mainly associated with ruling out the presence of a natural infection during the experiment that weak the interpretation of the results. The manuscript will considerably benefit if authors decipher whether natural infections happened during the experiment. Regardless of that, the information is valuable and exciting for the readers of the journals. My specific comments are listed below:

Major comments

A significant concern is a potential role that natural SARS-CoV-2 infection could have in everyone during the sample collection (Day 0-180). Is the information regarding close contact with SARS-CoV-2-positive people or clinical signs associated with SARS-CoV-2 during the sample collection available? How was the VOC circulation during the experiment timeline? Could this be a variable? Besides CoronaVac, the rest of the vaccines are Spike-based. Therefore, antibodies against N are a tool to differentiate natural infection and vaccination. That information will strongly support the idea of a lack of disease during the sample collection period. Anti-N antibodies are a piece of information already available to the authors based on the discussion.

An increasing body of literature demonstrates that biological sex plays a role in the immune response against vaccination and pathogenesis against respiratory viruses such as influenza or SARS-CoV-2. The authors successfully analyzed potential sex differences, finding such differences in the cohort analyzed (Line 121-23), which is highly informative in this reviewer's opinion. However, the authors failed to show these results appropriately, and just a table is in place. I strongly encourage the authors to present a figure based on sex differences.

Correlations with age were carried out. However, the biological sex was not analyzed.

Please discuss more in detail about potential explanations for the differences in the correlation between S1/S2 IgG vs. neutralization capacity between no prior covid and prior covid positive individuals.

Minor comments

Line 61-63: Please update the number of cases, deaths, and vaccines administered until September 2022 at least

Timeline of sample collection: Please specify specific numbers of individuals and the date of sample collection in materials and methods

Fig 1 will benefit from the use of a more precise nomenclature. Along the papers, the different samples are denoted by the day of collection, which I think should be the way to go in this figure.

Fig 3: Please specify each day on top or under each panel, so it's clear for the readers

Line 161: Please check the figures described. It should be 3C and 3G.

Fig 5 and 6. Please specify the name of each vaccine instead of A, B, C, etc.

Fig7: Please check the figure legend for C and D (is the same?)

Fig 8 and 9 should be combined.

Response to Reviewers

Reviewer 1

1. Study is designed well and gives good value to the current scenario. It would be useful and provide more value if this data can be correlated with the presence of a Variant of concerns or variants present during the study time. VOCs do affect the neutralization capacity of vaccine and their efficacy so if it's possible then incorporate or use other published data to describe the possible circulating VOCs.

Response: We included a supplementary figure that shows the VOCs that were circulating during the study period. In addition, we also included a new paragraph on the discussion.

Figure S2 shows VOC circulation during the study period, in the area where the samples were obtained. Our data only shows the percentage of neutralizing antibodies vs RBD, but not the neutralizing activity against the different variants. A limitation of our study was not performing neutralization assays against the different variants. Besides, since a large part of the Mexican population was not yet vaccinated during the study period, it is difficult to know if the waves occurred due to the lack of effectiveness of the vaccines used or due to the unvaccinated population.

In a study carried out by Bednarski et al, a reduction in neutralization was observed by Delta and Omicron variants measure by plaque neutralization assays. Even so, although many cases occurred in people vaccinated during the third (delta) and fourth waves (ómicron BA.1 and BA.2), these cases were generally milder and produced significantly fewer deaths, which indicates that even with the reduction of the neutralizing activity, the applied vaccines limited the evolution to severe stages of the disease.

2. Also how you have taken care of the individuals who were infected earlier with COVID and vaccinated or vaccinated and later got infected? Did this impact the outcome?.

Response: Due to the large number of asymptomatic infections, we decided to check whether the participants had already had the disease before vaccination by measuring antibodies against SARS-CoV-2 before immunization with the vaccines studied. Therefore, samples were taken from all the individuals studied minutes before vaccination (day 0), and thus, we were able to identify those who previously had COVID-19 (symptomatic or asymptomatic), from those who had never had contact with the SARS-CoV-2.

Our results show that participants who already had COVID-19 (identified with our assays) generated a much higher number of antibodies compared to the group that did not had COVID before.

Unfortunately, due to lack of resources, we were not able to measure antibodies against N, except for CoronaVac, which would give us information on the participants who had the disease during the study period, therefore, after vaccination. So, we couldn't know what was the impact of getting infected after immunization.

3. Did study was followed post-180 days to see the impact of breakthrough, re-infection or other impacts

Response: Yes, we analyzed the participants also at 270 and 360 days post-vaccination, even with the information on the heterologous booster vaccination, however, on the date of submission of this manuscript, we still did not have the analysis of these data and we are preparing for the publication of a second report, in which we will focus the analysis on boosters and heterologous vaccination. Unfortunately, as mentioned in the previous question, it was not possible to identify the subjects who became infected after the start of the study.

4. Was booster dose implemented in a later phase

Response: Yes, and in most cases the third dose was a different brand of vaccine. This information was not sent, because until the date of sending this manuscript, we still did not have these data, and as we mentioned before, we are still preparing these data for a later report.

5. How many breakthrough cases were reported during the study

Response: During the study, very few participants reported symptoms similar to those of COVID-19, but we do not have confirmatory data on whether or not they had the disease. With the intention of verifying if the participants had the infection during the follow-up, we should analyze the different samples in search of antibodies against the N protein (not produced with the vaccine, except for CoronaVac). Nevertheless, because of lack of resources does studies couldn't be done.

6. Those did not develop antibody, have breakthroughs in follow-up period of 6 months or not.

Response: We could not confirm that they had an infection during the study. However, we know that they received one or two doses of the vaccine (depending on the scheme), and that they were also exposed to 3 waves in the country, caused by different variants. Still,

there were participants who never developed antibodies, measured by the techniques used in this study.

7. Was comorbid group show any significant difference with the healthy group for antibody or NAb difference

Response: Following the reviewer's recommendation, differences in the generation of anti-S1/S2 and RBD antibodies in participants with and without comorbidities were analyzed in more detail. In the group without previous COVID-19, there were no significant differences in terms of the anti-S1/S2 IgG antibody response, nor in the percentage of inhibition in any of the comorbidities analyzed.

However, in the group with previous COVID-19, the people who suffered from arterial hypertension had a higher average of antibodies compared to those who did not suffer from it, a difference that appeared in the first shots, and that was no longer significant. at 180 days of follow-up.

A new paragraph was placed in the discussion section in this regard, and a supplemental table (Supplemental Table 2) was added.

8. Was this neutralization enzyme-linked immunosorbent assay (ELISA) specific to measure the neutralizing capacity of the anti-RBD antibodies for any specific VOCs??

Response: The initial approach aimed to determine the longevity of the antibodies generated by the different vaccines, which were designed based on the wildtype variant, for which ELISA and Neutralization tests designed against the original strain were selected. However, this does not necessarily mean that they do not detect antibodies produced from infection with other variants.

9. Limitation of the study should be expressed

Response: The study has 3 important limitations: the first is that for economic reasons it was not possible to determine the presence of antibodies against the N protein, and consequently we could not define infections during follow-up. The second is that plaque reduction neutralization test was not carried out to determine if the neutralizing capacity of the antibodies generated decreases in the face of the different variants, and finally the loss of participants during follow-up.

The paragraph that talks about this in the discussion was edited to make it clearer.

Reviewer 2

1. A significant concern is a potential role that natural SARS-CoV-2 infection could have in everyone during the sample collection (Day 0-180). Is the information regarding close contact with SARS-CoV-2-positive people or clinical signs associated with SARS-CoV-2 during the sample collection available?

Response: We have no information about the close contact of the participants with people with SARS-CoV-2 during the study. The determination of IgG antibodies against the N protein of the virus would have allowed us to better visualize this situation, however, we had economic limitations that did not allow us to do so. Even so, we decided to compare between the groups with and without previous infection, in which the probability of exposition is the same.

2. How was the VOC circulation during the experiment timeline?. Could this be a variable?

Response: We included a supplementary figure 2 that shows the VOCs that were circulating during the study period. In addition we also included a new paragraph on the discussion.

Figure S2 shows VOC circulation during the study period, in the area where the samples were obtained. Our data only shows the percentage of neutralizing antibodies vs RBD, but not the neutralizing activity against the different variants. A limitation of our study was not performing neutralization assays against the different variants. Besides, since a large part of the Mexican population was not yet vaccinated during the study period, it is difficult to know if the waves occurred due to the lack of effectiveness of the vaccines used or due to the unvaccinated population.

In a study carried out by Bednarski et al, a reduction in neutralization was observed by Delta and Omicron variants measure by plaque neutralization assays. Even so, although many cases occurred in people vaccinated during the third (delta) and fourth waves (ómicron BA.1 and BA.2), these cases were generally milder and produced significantly fewer deaths, which indicates that even with the reduction of the neutralizing activity, the applied vaccines limited the evolution to severe stages of the disease.

3. Besides CoronaVac, the rest of the vaccines are Spike-based. Therefore, antibodies against N are a tool to differentiate natural infection and vaccination. That information will strongly support the idea of a lack of disease during the sample collection period. Anti-N antibodies are a piece of

information already available to the authors based on the discussion.

Response: We agree with the observation, however, we only had access to a small number of reactions against the N protein, so we decided to use them in CoronaVac, since, as we commented in the article, we wanted to verify the data obtained against the S protein in that vaccine in specific, because the individuals immunized with this biological had a different behavior with respect to those who received the other vaccines

4. An increasing body of literature demonstrates that biological sex plays a role in the immune response against vaccination and pathogenesis against respiratory viruses such as influenza or SARS-CoV-2. The authors successfully analyzed potential sex differences, finding such differences in the cohort analyzed (Line 121-23), which is highly informative in this reviewer's opinion. However, the authors failed to show these results appropriately, and just a table is in place. I strongly encourage the authors to present a figure based on sex differences.

Response: In our study, the differences observed between the male and female sex were present only at the first contact with the antigen, so they had only been placed in Table 2. With the intention of making this result more visible and following the reviewer's recommendation, we have included a new supplementary figure (Figure S1)

5. Correlations with age were carried out. However, the biological sex was not analyzed.

Response: Comparative analyzes of the amount of anti-S1/S2 IgG antibodies and percentage of inhibition were carried out for both sexes among the participants of the different vaccines, but only sporadic differences were found, which seemed more like random behaviors derived from biological variability, than a real tendency derived from sex. The only pattern that really appears to be sex-linked is the response generated on first contact with the antigen, which was placed in Table 2, and now also in Supplementary Fig. 1.

6. Please discuss more in detail about potential explanations for the differences in the correlation between S1/S2 IgG vs. neutralization capacity between no prior covid and prior covid positive individuals.

Response: The following paragraph was added to the discussion section

The correlation analysis between the amount of IgG produced and the neutralization capacity showed that, contrary to what was expected, these follow a logarithmic relationship and not a linear relationship, which means that, even at the point where a person produces few anti-S1/S2 IgG antibodies, there will not necessarily be a low capacity to neutralize the

virus. A clear example that even with a low amount of anti-S1/S2 IgG antibodies, high levels of neutralization can be achieved, is what is observed in the group vaccinated with BNT162b2 after the first immunization. In this group, the logarithmic ratio was lower than the groups vaccinated with other brands, this because more people reached high levels of neutralization, despite the low amount of anti-S1/S2 IgG antibodies they produced in general”.

This finding becomes relevant to the interpretation of other works in which population immunity is measured based on kits directed against protein S and not specifically on the ability to neutralize the virus.

7. Conclusions should state better the categorization of the vaccines based on the experiments carried out.

Response: The conclusion section was modified

Our results showed that, from the first immunization, the analyzed vaccines caused different levels of seroconversion in the population (without previous infection), whose percentages varied from 54.2-87.8%, CoronaVac being the vaccine with the lowest and BNT162b2 with the highest percentage of seroconversion. After the second immunization, all the vaccines caused seroconversion levels above 90% (except Convidecia, which had a single-dose schedule). Despite the fact that the seroconversion percentages obtained in this study were lower than those reported in other parts of the world, given the current global scenario, where a good part of the population has already had a natural SARS-CoV-2 infection, the application of any of the biologicals analyzed in this study manages, in most cases, to induce a good production of anti-S1/S2 and neutralizing IgG antibodies, so they could be an option as a booster dose, as long as take care of the compatibility with the biological applied initially.

The fact that there is a percentage of the population in which there was no production of antibodies during the 180 days of the study leaves open some questions about the effectiveness of the vaccines themselves, the protection generated by the cellular immune response, the form of application, the cold network used to store them and the associated human factor. All these factors must be taken care of and investigated, to avoid the appearance of serious cases of COVID-19.

Minor comments

Line 61-63: Please update the number of cases, deaths, and vaccines administered until September 2022 at least

Response: The data was updated until the month of October.

Timeline of sample collection: Please specify specific numbers of individuals and the date of sample collection in materials and methods.

Response: In addition to Figure 9, the number of individuals for the analysis of each vaccine and their sampling date were added in text form.

Fig 1 will benefit from the use of a more precise nomenclature. Along the papers, the different samples are denoted by the day of collection, which I think should be the way to go in this figure.

Response: Correction was made.

Fig 3: Please specify each day on top or under each panel, so it's clear for the readers.

Response: Because the data in the graphs occupies almost the entire panel, we decided to modify the figure caption to make the information of the days clearer.

Line 161: Please check the figures described. It should be 3C and 3G.

Response: Correction was made.

Fig 5 and 6. Please specify the name of each vaccine instead of A, B, C, etc.

Response: The figure caption was modified as per reviewer's suggestion:

"... Lines A (BNT162b2), B (AZD1222), C (Convitecia), D (Sputnik V) and E (CoronaVac)."

Fig7: Please check the figure legend for C and D (is the same?).

Response: Correction was made.

Fig 8 and 9 should be combined.

Response: Correction was made.

December 6, 2022

Dr. Jose Esteban Muñoz-Medina
Instituto Mexicano del Seguro Social
Laboratorio Central de Epidemiología
Mexico
Mexico

Re: Spectrum02376-22R1 (Longevity and neutralizing capacity of IgG antibodies against SARS-CoV-2 generated by the application of BNT162b2, AZD1222, Convidecia, Sputnik V and CoronaVac vaccines: A cohort study in Mexican population.)

Dear Dr. Jose Esteban Muñoz-Medina:

Thankyou for addressing the reviewers' concerns. Your manuscript has been accepted, and I am forwarding it to the ASM Journals Department for publication. You will be notified when your proofs are ready to be viewed.

Sincerely,

Alison Sinclair
Editor, Microbiology Spectrum

Journals Department
Supplemental Material: Accept
Supplemental Dataset: Accept